# Nanoarchitectonics and Kinetics Insights into Fluoride Removal from Drinking Water Using Magnetic Tea Biochar

**DOI:** 10.3390/ijerph192013092

**Published:** 2022-10-12

**Authors:** Imtiaz Ashraf, Rong Li, Bin Chen, Nadhir Al-Ansari, Muhammad Rizwan Aslam, Adnan Raza Altaf, Ahmed Elbeltagi

**Affiliations:** 1School of Chemical Engineering, Northwest University, Xi’an 710069, China; 2Department of Civil, Environmental and Natural Resources Engineering, Luleå University of Technology, 97187 Luleå, Sweden; 3College of Environmental and Resource Sciences, Zhejiang University, Hangzhou 310027, China; 4College of Engineering, Huazhong Agricultural University, Wuhan 430070, China; 5Agricultural Engineering Department, Faculty of Agriculture, Mansoura University, Mansoura 35516, Egypt

**Keywords:** waste tea feedstock, magnetic sorbent, defluoridation, water treatment

## Abstract

Fluoride contamination in water is a key problem facing the world, leading to health problems such as dental and skeletal fluorosis. So, we used low-cost multifunctional tea biochar (TBC) and magnetic tea biochar (MTBC) prepared by facile one-step pyrolysis of waste tea leaves. The TBC and MTBC were characterized by XRD, SEM, FTIR, and VSM. Both TBC and MTBC contain high carbon contents of 63.45 and 63.75%, respectively. The surface area of MTBC (115.65 m^2^/g) was higher than TBC (81.64 m^2^/g). The modified biochar MTBC was further used to remediate the fluoride-contaminated water. The fluoride adsorption testing was conducted using the batch method at 298, 308, and 318 K. The maximum fluoride removal efficiency (*E*%) using MTBC was 98% when the adsorbent dosage was 0.5 g/L and the fluoride concentration was 50 mg/L. The experiment data for fluoride adsorption on MTBC best fit the pseudo 2nd order, rather than the pseudo 1st order. In addition, the intraparticle diffusion model predicts the boundary diffusion. Langmuir, Freundlich, Temkin, and Dubnin–Radushkevich isotherm models were fitted to explain the fluoride adsorption on MTBC. The Langmuir adsorption capacity of MTBC = 18.78 mg/g was recorded at 298 K and decreased as the temperature increased. The MTBC biochar was reused in ten cycles, and the *E*% was still 85%. The obtained biochar with a large pore size and high removal efficiency may be an effective and low-cost adsorbent for treating fluoride-containing water.

## 1. Introduction

Fluoride is a naturally occurring element that can be exposed to humans through food and drinking water. The health effects of fluoride are mainly concentration dependent, as both low and high concentrations cause serious harm [1]. The WHO (World Health Organization) recommended the fluoride concentration of 0.5–1.5 mg/L in drinking water [2]. A low fluoride level (<0.5 mg/L) helps to treat dental decay and osteoporosis in the human body. Therefore, it is allowed to be added into drinking water, pharmaceutical agents, and toothpastes [3]. In contrast, a high concentration of fluoride (>1.5 mg/L) causes several health problems, such as skeletal and dental fluorosis, sweating, restlessness, and joint deformation [4]. A high fluoride concentration also causes neurotoxicity in children, which results in lower IQ (intelligent quotient) levels [3]. Excessive fluoride in drinking water is due to the combination of anthropogenic activities, weathering of fluorine minerals, and industrial activities [5]. Pakistan [6], China [7], India [8], and the rift valley regions of African countries are the most affected areas [9]. In Pakistan, the drinking water of most areas is enriched with fluoride [10]. Small size, high charge density, capabilty of forming H-bonding, and strong basic nature makes its detection tedious [11]. Varios methods, such as electrochemical [3], spectroscopic [12] and fluorescent sensors [13], and chemosensors [14], are used for fluoride detection. Excessive fluoride concentration in drinking water is an important environmental problem worldwide, due to its negative and toxic effects [5]. Therefore, fluoride removal from drinking water is essential for human health.

Various methods, including ion-exchange [15], reverse-osmosis [16], nano-filtration [17], co-precipitation [18], electrodialysis [19], and adsorption [20], have been developed for water defluoridation. The technology is used according to the actual conditions of the water treatment. Adsorption is the most commonly used method due to its high mobility, low price, high proficiency, and environmentally friendly property. Therefore, high-quality adsorbents with excellent properties attract more attention. Many adsorbents, such as activated carbon [21], bone char [22], zeolite [23], calcined clay [24], activated fly ash [25] and loaded biochar [26], have been used for water defluoridation. Recently, biochar (BC) has been extensively used as an adsorbent due to its low-budget, eco-friendly nature, and substantial resources [27]. It is a porous carbon raw material with high S_BET_, functional groups, and is suitable for adsorbing various environmental pollutants [27].

In our environment, wastes are generated from different sectors on a daily basis and the agricultural sector produces large amounts of waste. Dietary fiber, animal manure, risk husk, banana peels, coconut shell, and tea are the important wastes that belong to the agricultural sector [28]. Tea is the most widely used beverage, with nearly 3.5 million tons of tea being used per year globally [27]. More than 90% of tea leaves are left over after consumption in restaurants, teashops, and homes, which raises serious waste disposal concerns [27]. This waste provides a renewable resource for biochar production. Furthermore, tea contains different biochemical parts, such as cellulose, hemicellulose, lignin protein, and tannins [29]. Different groups, such as phenolic, carboxylic, aromatic, and oxyl/hydroxyl groups, are present in these biochemical parts, which help to remediate the toxic elements [30]. Therefore, using tea waste for biochar production might be a feasible approach for waste administration and renewable material for defluoridation of groundwater [31].

Most of the biochar research mainly focuses on its improvement, high performance, and carbon-storing. In the past ten years, the use of biochar to treat groundwater has gained more attention worldwide [4]. It is difficult to separate the biochar from the natural source, which may produce further contamination and affects the scope of biochar as an adsorbent [32]. In this study, magnetic tea biochar is prepared, which can be removed easily after the adsorption process [33]. Literature data on the influence of magnetization indicate that different reactions can be observed on biochar’s surface that increase or decrease the adsorption capacity [27]. The application of Fe affects the biochar’s adsorption capacity and selectivity due to its surface modification, leading to “engineered chemical biochar”. In previous work, (ZnCl_2_ and FeCl_3_) and (K_2_CO_3_, and Fe_3_O_4_) combinations are used as magnetic precursors [34,35]. However, using these magnetic precursors may be limited by high preparation temperature, high micro-porosity in carbon, corrosion, and the high mass ratio of ZnCl^2^ and K_2_CO_3_ [34,35]. Recently, Fe^3^O_4_ (magnetite) and Fe_2_O_3_ (maghemite) have attracted worldwide attention as they have high redox potential, a large surface area, and are a cheap magnetic source. Fe (NO_3_)_3_·9H_2_O is a precursor for Fe_3_O_4_/γ-Fe_2_O_3_ as it is non-toxic and an oxidant in chemical reactions [36].

In this study, magnetic tea biochar (MTBC) was prepared by Fe (NO_3_)_3_·9H_2_O-loaded waste tea leaves to remove the fluoride. Surface area, phase difference, morphology, magnetic properties, and chemical bond of the adsorbent before and after fluoride adsorption were investigated. Initial fluoride concentration, pH value, adsorbent dosage, adsorption temperature and time, and the influence of competing anions on fluoride removal efficiency were measured. Regeneration study and real water defluoridation for this adsorbent have been investigated, along with adsorption isotherm, kinetics, and thermodynamics study.

## 2. Material and Methods

### 2.1. Chemical and Reagents

Tea was purchased from the local market. In order to reproduce the tea waste, tea leaves were soaked in hot water and after every hour, fresh water was replaced with soaked water and this continued for six hours. This tea waste was covered with a net and left to dry for 24 h under sunlight. Tea leaves were crushed and sieved to obtain 120–200 mesh size and preserved for the subsequent process.

All chemicals, including Fe(NO_3_)_3_·9H_2_O (ferrous nitrate), HCl (hydrochloric acid), NaOH (sodium hydroxide), NaF (sodium fluoride), C_6_H_9_Na_3_O_9_ (sodium citrate dihydrate), NaHCO_3_ (sodium bicarbonate), Na_2_CO_3_ (sodium carbonate), NaCl (sodium chloride), Na_2_SO_4_ (sodium sulphate), and NaNO_3_ (sodium nitrate), are analytical grades with no further purification. First, 2.210 g of NaF was added to 1.0 L deionized water to form a 1000 mg/L fluoride stock solution, and other fluoride concentrations were obtained by dilution. HCl and NaOH (0.1 M) solutions were used for pH adjustment.

In this study, we used an Orion ion-selective electrode (OISE) to determine the F^−^ concentration in the solution (Thermo Fisher Star-A214, Waltham, WA, USA). This OISE is a type of chemical sensor that works based on the principle of galvanic cells. The F^−^ electrodes were calibrated for 10, 100, and 500 mg/L concentrations, and total ionic strength was adjusted with a buffer (TISAB). The calibration electrode was used to measure the F^−^ concentration of all the experimental water samples [4].

### 2.2. Sorbent Preparation

Household tea waste was used as a precursor of sorbent preparation, as described in our previous work [27]. First, 10 g of tea waste was dipped in the ferrous nitrate solution (0.46 M) for Fe impregnation, stirred continuously at 300 rpm at 60 °C, and dried at 105 °C for 12 h. Then, pyrolysis was carried out in a quartz reactor equipped with a PID temperature controller and labeled as MTBC [37]. The BT biochar was prepared using the same pyrolysis method without Fe impregnation and labelled as TBC. Finally, both sorbents were kept in a sealed flask for further analysis and usage.

### 2.3. Characterization

#### 2.3.1. Ultimate and Proximate Analysis

The TBC and MTBC were heated at 105 °C for *MC* (moisture content, ASTM-D2867-09) and at 750 °C for *AC* (ash content, ASTM-D2866-11). Furthermore, *VC* (volatile content) was determined by using the standard method (ASTM-D2866-11). *FC* (fixed carbon content) was calculated by using Equation (1).
(1)$$FC=\left[100-\left(VC+AC+MC\right)\right]\;$$

Carbon, hydrogen and nitrogen contents were determined by using an elemental analyzer (EA3000, Euro-Vector). In contrast, the *O* content was determined by using Equation (2).
(2)$$\%\;O=\left[100-\left(C+H+N\right)\right]\;$$

#### 2.3.2. Surface Area, Morphology, Magnetic and Thermal Analysis

BET (Brunner Emmet Teller-autosorbic2) was used to determine the total surface area, pore size, and pore volume. SEM (scanning electron microscopy, Joel 7100F) was used to determine the surface morphology before and after adsorption. XRD (X-ray diffractometer, Phillips APD3720) was used to determine the adsorbent phase before and after adsorption. FTIR (Fourier transform infrared spectrometer, Prestige FT-IR8400s) was used to identify the interaction between the adsorbent and fluoride ions. VSM (vibrating sample magnetometer) was used to analyze the magnetic properties of the prepared sorbent. TGA (Thermogravimetric system-Q500) was used to measure the thermal stability. For TGA, 10 mg of TBC and MTBC was taken and heated from 30 to 750 °C under N_2_ flow of 100 mL min^−1^ at 20 min^−1^ °C.

#### 2.3.3. Point of Zero Charge Analysis

The salt addition method was used to determine the pH values of the adsorbent at point zero charges (pH_PZC_). Initially, 0.02 g of the adsorbent was placed in a 50 mL Erlenmeyer flask containing 40 mL of NaNO_3_ (0.1 M) solution. The solution pH that ranged from 2.6 to 12.6 was changed by 0.1 M HCl and 0.1 M NaOH solution. Preliminary experiments confirmed no change in the pH after about 24 h. Therefore, the suspension was stirred with a shaker (Yamato, MK-200D) for 48 h in the air conditioning room (25 ± 1 °C) for stable equilibrium adsorption. After this, it was centrifuged for 15 min at 180 rpm and filtered. A pH meter was used to determine the pH of the supernatant. A graph was drawn between ΔpH, and the adsorbent’s initial pH value and the point of zero charges (pH_PZC_) were determined [38].

### 2.4. Adsorption Experiment

A batch method was used to calculate fluoride’s adsorptive capacity and removal efficiency of the adsorbent. Different parameters, such as adsorbent dosage, initial fluoride concentration, solution pH, contact time, and contact temperature, were optimized for adsorption. In this experiment, 50 mL fluoride solution was placed in a 100 mL conical flask using an incubator shaker with 150 rpm speed by varying the adsorbent dosage (0.1–1.0 g/L), initial fluoride concentration (1–500 mg/L), solution pH (2.6–12.6), contact time (1–24 h), and contact temperature (293–318 K). Equations (3) and (4) were used to calculate the fluoride’s adsorption capacity (*q_e_*) and removal efficiency (*E*%), respectively.
(3)$$q_{e}=\frac{\left(C_{i}-C_{e}\right)\times V}{m}\;$$
(4)$$E\;\%=\frac{\left(C_{i}-C_{e}\right)}{C_{i}}\times 100\;$$
where *C_i_* and *C_e_* represent the initial and equilibrium adsorbate concentration (mg/L), *V* is the volume of fluoride solution (L), and *m* is the mass of the adsorbent (g).

#### 2.4.1. Adsorption Kinetics Study

In this experiment, 0.5 g of adsorbent was added into a 50 mL fluoride solution with different concentrations (10, 30, 50, 100, 150, and 200 mg/L) in a 100 mL conical flask, using an incubator shaker with 150 rpm speed at 308 K. Fluoride concentration was determined by the Orion ion-selective electrode.

Furthermore, the obtained data were analyzed by different kinetic model such as pseudo first order (PS1), pseudo second order (PS2), Elovich and intraparticle diffusion model, to explain the adsorption on biochar^1^ by using Equations (5)–(10), respectively.
(5)$$ln\left(q_{e}-q_{t}\right)=lnq_{e}-k_{1}t\;$$
(6)$$\frac{t}{q_{t}}=\frac{1}{k_{2\;}q_{e}2}+\frac{1}{q_{e}}t\;$$
(7)$$h=k_{2}{q_{e}}^{2}$$
(8)$$q_{t}=\frac{1}{\alpha }ln\left(\alpha \beta \right)+\frac{1}{\alpha }lnt$$
(9)$$q_{t\;}=k_{d}t^{1/2}+C\;$$
(10)$$R_{i}=1-\frac{C}{q_{e}}\;$$
where *q_e_* and *q_t_* represent the adsorption capacity (mg/g) at equilibrium and time *t*. *k_1_* (1/h) and *k_2_* (g/mg·h) represent the pseudo 1st and 2nd order rate constant. *h* shows the initial rate (mg/g·h). *α* and *β* represent the adsorption and desorption rate constant (g/mg·h). *k_d_* is the intraparticle diffusion constant (mg/g·h^1/2^). *C* represents the characteristic parameter of intraparticle diffusion, and *R_i_* is the initial adsorption factor.

#### 2.4.2. Adsorption Isotherm Study

In this experiment, 0.5 g of adsorbent was added into a 50 mL fluoride solution with different concentrations (10, 30, 50, 100, 250 and 500 mg/L) in a 100 mL conical flask, using an incubator shaker with 150 rpm speed at different temperatures (298, 308, and 318 K). Fluoride concentration was determined by the Orion ion-selective electrode.

Moreover, the fluoride adsorption data were analyzed by different isotherm models, including the Langmuir, Freundlich, Temkin and Dubnin–Radushkevich model, to explain the fluoride adsorption mechanism on the biochar by using Equations (11)–(15), respectively.
(11)$$\frac{C_{e}}{q_{e}}=\frac{1}{q_{m}}C_{e}+\frac{1}{K_{L}q_{m}}\;$$
(12)$$logq_{e}=logK_{F\;}+\frac{1}{n}\;logC_{e}$$
(13)$$q_{e}=\frac{RT}{b}lnK_{T}+\frac{RT}{b}lnC_{e}$$
(14)$$q_{e}=q_{mDR}exp\left(-b_{DR}{\left(\ln\left(1+\frac{1}{C_{e}}\right)\right)}^{2}\right)$$
(15)$$E=\frac{1}{\sqrt{2B_{DR}}}\;B_{DR}=\frac{b_{DR}}{{\left(RT\right)}^{2}}$$
where *C_e_* represents the final adsorbate concentration (mg/L) and *q_e_* represents the equilibrium adsorption capacity (mg/g). *q_m_* represents the maximum monolayer adsorption capacity (mg/g) related to their isotherms, and *K_L_* is the Langmuir constant (L/mg). *K_F_* and *n* show the adsorption capacity (mg/g) and adsorption intensity (L/mg)^n^. *E* is the adsorption energy (kJ mol^−1^), *R* is the general gas constant (8.314 kJ/K/mol), and *T* is the absolute temperature (K). *b* and *K_T_* represent the constant heat for the Temkin isotherm (kJ mol^−1^) and equilibrium binding constant (g/mg).

### 2.5. Effect of Co-Exist Ions

Actual groundwater contains many other anions, which affect the adsorption capacity [39]. So, it is important to show the performance of the synthesized adsorbent in the presence of these ions (SO_4_^2^^−^, Cl^−^, NO_3_^−^, CO_3_^2^^−^, and HCO_3_^−^). The initial fluoride concentration was 50 mg/L, and the adsorbent amount was 0.35 g at pH = 5.6 for 16 h contact time. The initial concentration of these anions varies from 10 to 100 mg/L, respectively, and can be used to find out the effect of these anions.

### 2.6. Regeneration and Real Water Test

The renewability of the adsorbent plays an important role in its industrial application. Thus, a regeneration study of MTBC was conducted after fluoride adsorption. In this study, the initial fluoride concentration and adsorbent amount were 50 mg/L and 0.35 g, respectively. Firstly, the used adsorbent was filtered and washed numerous times with deionized water. Secondly, the adsorbent was mixed with 50 mL 0.1 M NaOH solution and kept in a water shaker bath with 150 rpm at 30 °C for 3 h for desorption [6]. Ten regeneration cycles were performed to analyze the adsorbent reusability.

Some groundwater samples were collected from different parts of Lahore, Pakistan, for actual water treatment. Some of the water samples had fluoride concentrations that were lower than the permissible limit. So, 2 M NaF was added to increase the fluoride concentration to measure the adsorbent’s ability to remove the fluoride [6]. In this experiment, 140 mL + 10 mL (groundwater + 2 M NaF solution) samples were added to a 250 mL flask to determine the fluoride adsorption at 308 K (dose = 0.5 mg/L, pH = 3.6, time = 24 h).

## 3. Results and Discussion

### 3.1. Proximate, Ultimate and BET Analysis

The analysis of TBC and MTBC includes fixed carbon (*FC*), moisture content (*MC*), ash content (*AC*), and volatile matter (*VM*) and these are represented in Table 1. Additionally, the corresponding physical properties, such as surface area, pore volume, and average pore width, are given in Table 1. It was observed that the pore volume, pore width, and specific surface area were improved in MTBC rather than TBC. This indicates that Fe (NO_3_)_3_ modification supports the formation of pores during pyrolysis [28]. Figure 1 display the N_2_ adsorption–desorption graph of prepared adsorbents with their pore desorption distribution by BJH analysis (Barrett Joyner Halenda). Nitrogen capacity remained constant in the low-pressure zone, while it increased as the relative pressure reached 0.8, given the hysteresis loop in both cases. On the other hand, the pore width distributions in TBC and MTBC were in the range 2~6 and 5~15, respectively, which shows that MTBC and TBC have a mesoporous structure [6]. 

### 3.2. Morphology, Magnetic and Thermal Analysis

#### 3.2.1. SEM Analysis

Figure 2a–d represent the adsorbent surface’s morphology before and after adsorption. Initially, the tea waste surface without pyrolysis has a porous structure, as observed in Figure 2a. During the pyrolysis reaction, this porous structure changed into a mesoporous structure due to the increased production of volatile components, as observed in Figure 2b [28,33]. Figure 2c demonstrates that the Fe_3_O_4_ particles were formed and showed good dispersion on the MTBC’s surface, which is indicative of a high S_BET_. Furthermore, the attachment of Fe particles confirmed by the magnet is shown in Figure 5. After fluoride adsorption, more small particles are found on the adsorbent’s surface, as observed in Figure 2d.

#### 3.2.2. XRD Analysis

Figure 3 represents the XRD configuration for the prepared adsorbent before and after adsorption. TBC consists of porous carbon, which reacts with SiO_2_ to form SiC, and this peak was shown in the XRD pattern (JCPDS:43-0697) [33]. The broad peak was observed at 2θ = 24.36, indicating organic groups (lignin and cellulose), as tea ash have these components. The MTBC was mainly composed of Fe_3_O_4_ and Fe_2_O_3_ elements, whose primary reflection was shown to correspond to Fe_3_O_4_ (JCPDS card number 74-0748) and Fe_2_O_3_ (JCPDS card number 39-1346) [36]. After fluoride adsorption, the XRD pattern clearly showed that SiC, Fe_3_O_4_, and Fe_2_O_3_ react with fluoride ions to form SiF, FeF_3_, and FeF_2_ (JCPDS: 75-0097) [6] as their peaks’ intensity decreases [37].

#### 3.2.3. FTIR Analysis

The FTIR spectra shows the enrichment of adsorbent surfaces with different functional groups, which was helpful in increasing the adsorption capacity. Figure 4 shows the FTIR spectra of the prepared adsorbent materials before and after pyrolysis during Fe impregnation. Tea waste material has a broad peak at 3346 cm^−1^ due to the –OH polar group. Peaks at 2942 and 2890 cm^−1^ indicate the presence of CH_3_ in the aromatic ring of lignin and cellulose. Similarly, the peak at 1635 cm^−1^ is due to C=O and C=C vibration, indicating complex carbon components. The peaks at 1448 and 1242 cm^−1^ are due to –CH_2_ and CO– aromatic and OH– phenolic groups, respectively. The peak at 1048 cm^−1^ is due to C–O–C aliphatic and –OH alcoholic bonds, representing the oxygen-containing compound in cellulosic and ligneous components [38]. After pyrolysis and Fe impregnation, the reduced intensity and shifting of peaks at 3352, 1643, 1452, 1244, and 1042 cm^−1^ are due to the effect of water and high temperature during the pyrolysis reaction. The new peak appears at 561 cm^−1^, which indicates the presence of Fe–O in Fe_3_O_4_ [28]. After fluoride adsorption, the intensity of the peaks remains the same, and a new peak at 979 cm^−1^ indicates the C–F stretching [26]. In addition, the characteristic peak shift at 592 cm^−1^ indicates the Fe–F bond formation. The characteristic peak at 3359 cm^−1^ becomes broader and more robust, indicating the formation of hydrogen bonding [36]. It has been observed that oxygen-containing compounds play an essential role in the adsorption process.

#### 3.2.4. VSM Analysis

Magnetic properties are a crucial characteristic of synthesized sorbents, which likely made their separation from fly ash possible [27]. In Figure 5, the magnetic hysteresis curve is displayed, which indicates that the Fe-modified biochar shows good magnetic properties with small coercivity and negligible residual magnetization. This VSM analysis also matches with the XRD analysis, in which many Fe_2_O_3_ and Fe_3_O_4_ peaks are observed in the spectra. This VSM analysis also shows that the magnetization of MTBC after 10 consecutive reused cycles has been reduced to only 9% due to the formation of the Fe–F bond after adsorption.

#### 3.2.5. Thermogravimetric Analysis

Thermal properties were analyzed to determine the difference in fluoride removal of MTBC and TBC. The three stages of mass degradation in TBC are shown in Figure 6a. Initially, the Δm loss is 4.11% due to water evaporation at 80 °C [27]. The second mass loss was recorded between 200 and 400 °C. The Δm loss is 27.65% due to the degradation of hemicellulose, cellulose, and lignin in tea [33]. Above 500 °C, the mass–loss rate slows down and eventually reaches 800 °C, with a residual weight of 56.1%. While MTBC shows different TG (Figure 6a) and DTG (Figure 6b), the Δm loss rate is relatively lower than that of the original sample. Figure 6a shows a slight Δm loss of 2.15% at 80 °C. It is due to Fe binding on the sample surface to unbound water. However, 9.15% of the weight loss was due to the gas produced through a rapid decomposition process at 300–385 °C. Similarly, at high temperatures (500 to 750 °C), the instantaneous mass decomposition and gaseous product generation rate slows down, reaching 71.85% residual weight. Therefore, the slight Δm loss of the Fe-modified sample specifies that a very steady cross-linked product may be formed during the pyrolysis process. This prevents further oxidation/decomposition at high temperatures [37]. Therefore, Fe modification increases thermal stability and also provides the effective operating temperature of the MTBC material. This temperature range should be used in subsequent thermodynamic studies at 298, 308, and 318 K. In other words, the purpose of thermogravimetric analysis is not only to prove that the thermal stability of MTBC is better than that of TBC, but also provide the effective operating temperature range of MTBC.

### 3.3. Batch Adsorption Experiments

#### 3.3.1. Effect of pH and Point to Zero Charges (pHpzc) Analysis

The pH of the fluoride solution plays an important role in the charge surface and dynamic sites of the adsorbent [40]. Furthermore, protonation and deprotonation of various functional groups, such as amino, hydroxyl, and fluoride ionic forms, are affected strongly by solution pH [41]. As shown in Figure 7a, the fluoride adsorption capacity decreased as pH of the solution increased and sharply decreased when pH > 10.6, which was due to the release of OH^−^. At high pH, the OH^−^ ions compete with F^−^ for the active place, which leads to the drop in fluoride adsorption [40]. Furthermore, the maximum *q_e_* (mg/g) was observed at pH 3.6, which declines when the pH decreases from 3.6 to 2.6. This is due to HF formation, which mainly decreases the free fluoride ions at this pH (pK_a_ value of HF is 3.2) [42]. Furthermore, these results were consistent with the ZP analysis, as shown in Figure 7b. The ZP value (pH_pzc_) of MTBC was 5.6. It is thought that at pH < 5.6, the adsorbent surface is positive and has the attraction force, which increases fluoride *q_e_* (mg/g). In contrast, at pH > 5.6, the adsorbent surface becomes negative and has repulsion forces, which decreases the fluoride *q_e_* (mg/g) [43]. From Figure 7a, it can be noticed that three types of attraction are present during the process. At pH 0–5.6, the force of attraction is responsible for fluoride adsorption. At pH 5.6–10.6, the force of repulsion and anion exchange was responsible for fluoride desorption. At pH > 10.6, the strong competition of OH^−^ is responsible for fluoride desorption [44].

#### 3.3.2. Effect of Adsorbent Dosage

The fluoride adsorption capacity is simultaneously affected by the adsorbent dosage. This was studied by changing the adsorbent amount from 0.1 to 1.0 g/L, while the initial fluoride concentration (50 mg/L), pH (3.6), contact time (24 h), and temperature (308 K) are optimized values in this experiment. Appendix A represents the fluoride adsorption capacities and removal efficiency at the different adsorbent amounts. It was observed that initially, fluoride removal efficiency increased as the adsorbent amount increased from 0.1 to 0.5 g (52.8 to 97.6%). When the adsorbent amount was increased from 0.5 to 1.0 g, the uptake capacity slightly decreased due to limited adsorption sites. For a constant adsorbent amount, the total sites are limited. Therefore, at a higher amount, there will be fierce competition between the adsorbents for fluoride adsorption, which slightly decreases the adsorption efficiency [45].

#### 3.3.3. Effect of Initial Fluoride Concentration

The initial solution concentration also plays an essential role in increasing or decreasing the adsorption capacity. For this, the initial fluoride concentration varies from 1 to 500 mg/L at optimized values (the amount of adsorbent is 0.5 g/L, the adsorption time is 24 h, the adsorption temperature is 308 K, pH of the fluoride solution is 3.6, and solution volume is 50 mL). Appendix A represents the fluoride adsorption capacities and removal efficiency at different initial fluoride concentrations. Initially, the fluoride removal efficiency increased from 68 to 98% when the initial fluoride concentration varied from 1 to 50 mg/L. After this, the *E*% decreased by further increasing the initial fluoride concentration. This is due to the low fluoride concentration at the beginning and the sufficient surface area of the adsorbent, meaning that the adsorption rate was independent from the initial concentration. At moderate initial concentration, the driving force for fluoride that reached the active sites of adsorbent surface was enhanced, and the increased adsorption opportunities improve the adsorption efficiency. However, if the available sites on the adsorbent surface become saturated, the fluoride removal efficiency would decrease [46].

### 3.4. Adsorption Kinetics Study

Kinetics study usually help us to understand the adsorption mechanism of fluoride on MTBC surfaces. Four kinetics models are shown in Figure 8, which described the adsorption mechanism. The PSI model assumes that physical diffusion is the main control factor for the adsorption rate, while the PS2 model assumes that chemical adsorption is the main control factor for the adsorption rate. Comparing the linear correlation coefficient between PS1, PS2, and the Elovich model, the PS2 has the highest value, and the r_2_^2^ is above 0.999 (shown in Table 2). The *q_e,cal_* values that are constant with the *q_e,exp_* values indicate that PS2 well described the adsorption process on MTBC. PS2 assumes that the binding interaction between fluoride and the adsorbents was not completely physical, but indicated chemical interaction [1,7]. Furthermore, the r_1_^2^ values for the intraparticle diffusion model were above 0.95, indicating that the adsorption on the MTBC surface was controlled by both chemical and intraparticle diffusion.

The linear plot of *qt* vs. t ½ passing through the origin specified that intraparticle diffusion is the only rate-controlling step. Otherwise, other measures control the adsorption process [1]. In this study, the linear plot of *qt* vs. t ½ did not pass through the origin, so the multi-linearity of fluoride adsorption on MTBC was mainly described by two steps, as shown in Figure 8. The slope of each line described the adsorption rate, and higher values mean faster adsorption, and C described the thickness of the boundary layer [47]. Table 2 shows the *k_d1_* and *k_d2_*_,_ and C values. It can be observed that *k_d1_* > *k_d2_* for all concentrations, signifying that boundary diffusion is more suitable for fluoride adsorption monitored by intraparticle diffusion. Furthermore, as shown in Table 3, the Ri values indicate the initial adsorption process, and the results indicate that the initial adsorption can be described as intermediate for all concentrations. From the above discussion, we concluded that both PS2 and intraparticle diffusion models described the adsorption of F^−^ on the MTBC surface.

### 3.5. Adsorption Isotherms Study

The increase in fluoride adsorption capacities is mainly due to the high driving forces between the adsorbent and adsorbate, which was indicated by the isotherm study [25]. Figure 9 shows the evaluated results using the four isotherm models (Langmuir, Freundlich, Temkin, and Dubnin–Radushkevich) and related parameters, as shown in Table 4. The Langmuir isotherm defined monolayer adsorption, whereas the Freundlich isotherm defined the heterogeneous layer formation on the adsorbent surface due to adsorption. The Temkin isotherm described the adsorbent–adsorbate interaction. From these isotherms, it can be observed that the Freundlich model well described the adsorption on MTBC, as the regression coefficient values (r^2^ > 0.99) were higher than the regression coefficient values for the Langmuir model (r^2^ > 0.95). The *K_L_*, *K_F_*, and *q_m_* values decreased as the temperature increased, indicating that a low temperature is favorable for adsorption. The *R_L_* values described the feasibility of the adsorption process. For this, the adsorption condition is unfavorable for *R_L_* > 1. The adsorption condition is favorable for 0 < *R_L_* < 1. For this study, the *R_L_* values vary from 0.89 to 0.91, which indicates that the fluoride adsorption on MTBC is favorable. The *n* values vary from 2.30 to 2.39. Generally, the *n* values from 1 to 10 are favorable for adsorption. The *E* values vary from 1.06 to 0.90, which is less than 8 kjolmol^−1^, implying that physisorption occurred [48].

Overall, it was observed from Table 4 that the multilayered process given by the Freundlich model controlled the fluoride adsorption on MTBC. The b values from the Temkin model decreased as the temperature increased, signifying the exothermic process during adsorption.

Appendix A presented the comparison of Langmuir adsorption capacities of the prepared adsorbent with other used adsorbents for fluoride adsorption. It seemed that the adsorption capacity of MTBC is better than the other adsorbents described in the literature. However, it is impossible to obtain any performance results regarding the adsorbent capacities, as these were calculated at different conditions, such as pH, temperature, adsorbent dosage, adsorbent concentration range, adsorbent particle size, and surface area. It is challenging to compare the financial possibility of biochar with the reported biochar due to the various preparation conditions, property uses, and adsorption capacity [49]. For biochar, low temperatures and constant contact time are more important because low temperatures and low heating rates more effectively improve the recovery rate of biochar [50].

### 3.6. Adsorption Thermodynamics Study

The adsorption mechanism during fluoride adsorption on MTBC was investigated by adsorption thermodynamics. For this, ∆*G*° (standard free energy), ∆*H*^o^ (standard enthalpy), and ∆*S*° (standard entropy) were computed from Equations (16)–(18).
(16)$$∆G^{{}^{\circ}}=-RTlnK_{c}\;$$
(17)$$lnK_{d}=\frac{∆S^{{}^{\circ}}\;}{R}-\frac{∆H^{{}^{\circ}}}{RT}\;$$
(18)$$∆G^{{}^{\circ}}=∆H^{{}^{\circ}}-T∆S^{{}^{\circ}}$$

*K_d_* can be calculated using *αq_e_/C_e_*^6^_,_ where *α* is the adsorbent concentration_,_
*R* (8.314 Jmol^−1^K^−1^) represents the general gas constant, and *T* is the absolute temperature (K). The ∆*H*^o^ and ∆*S*^o^ values were drawn from the *lnK_d_* vs. *1/T* plot (Appendix A) and displayed in Appendix A. It can be observed that ∆*H*^o^ < 0, indicating that the adsorption process is exothermic. The negative values of ∆*G*^o^ indicated that fluoride adsorption is a spontaneous process, and its value decreased as the temperature increased, indicating that a low temperature is favorable for adsorption. The ∆*S*^o^ < 0, indicating that the adsorption occurs in the solution on the adsorbent surface.

### 3.7. Effect of Co-Existing Anions

Groundwater contains several anions (Cl^−^, NO_3_^−^, CO_3_^2−^, SO_4_^2−^, and HCO_3_^−^), which affect the fluoride adsorption on MTBC. For this reason, we have tested the fluoride adsorption in the presence of these ions. From Figure 10, it can be observed that Cl^−^ and HCO_3_^−^ slightly decreased the removal efficiency from 98 to 91 and 94, respectively. With increasing the concentration, similar results were observed, which shows that these anions have no effect. In contrast, anions such as NO_3_^−^, CO_3_^2−^, and SO_4_^2−^ changed the removal efficiency of fluoride due to their high charge density, according to the HSAB concept [31]. These anions compete with F^−^ for active spots on the surface during the adsorption process, which decreased the fluoride adsorption. The fluoride adsorption has little effect as the concentrations of these anions increase.

### 3.8. Regeneration Study and Application of Groundwater Samples

Ten consecutive regeneration cycles have been used to analyze the adsorption–desorption of MTBC for F^−^. From Appendix A, it can be observed that the adsorption capacity of the regenerated adsorbent still achieved 85%. The fluoride adsorption on MTBC was 85% after ten recycling processes, while the initial adsorption capacity was 98%. It can be noticed that after the tenth consecutive use of MTBC, the adsorption efficiency reduced by 13%, so we suggest that this adsorbent is helpful and supportable in fluoride removal in commercial and industrial use.

Some groundwater samples in the summer season from different parts of Lahore, Pakistan, were collected for fluoride adsorption. Some physiochemical and fluoride concentrations are displayed in Appendix A. The mean values of water characteristics are displayed as pH (7.29), TDS (63.8), ECs (509 µS/cm), Na^+^ (314 mg/L), K^+^ (3.5 mg/L), Cl^−^ (20.63 mg/L), F^−^ (1.32 mg/L), NO_3_^−^ (1.39 mg/L), SO_4_^2−^ (39.3 mg/L), and HCO_3_^−^ (332 mg/L), respectively. Initially, 20% of the samples exceeded the WHO values, which did not help us to understand the adsorption performance of the adsorbent. So, we added NaF and the results are shown in Figure 11, which shows the fluoride levels in groundwater before and after adsorption. It seemed that after adsorption, all the groundwater samples had a low fluoride concentration of <1.5, which is the acceptable limit set by the WHO in groundwater. Thus, MTBC could be used to remove the F^−^ from groundwater in the presence of competitive anions.

### 3.9. Possible Removal Mechanisms

Many factors, such as contact time, adsorbent dosage, pH, initial fluoride concentration, kinetics study, and thermodynamics study, affect the possible mechanism of fluoride adsorption on the MTBC surface [51]. For example, the pH influenced the degree of ionization and speciation without changing the charge density on the adsorbent surface [52]. Fluoride is the dominant species at pH > 3.6, while HF or H_2_F_2_ species are more stereotypically predominant at pH < 3.60. Furthermore, this optimum pH of 3.6 shows the maximum removal efficiency because it is less than the pH_pzc_ value, suggesting a positively charged adsorbent surface. So, with these points in mind, the fluoride adsorption on MTBC follows the electrostatic interaction between the positively charged surface and F^−^ and ion-exchange mechanism [10]. The possible mechanism on the MTBC surface is given as follows. 

$$\mathrm{MB}–\mathrm{OH}\;+{\;H}^{+}\;↔\;\mathrm{MB}–{\mathrm{OH}}_{2}{}^{+}$$$$\mathrm{MB}–{\mathrm{OH}}_{2}{}^{+}+F^{-}↔\;\mathrm{MB}–{\mathrm{OH}}_{2}–F$$$$\mathrm{MB}–{\mathrm{OH}}_{2}{}^{+}+F^{-}\;↔\mathrm{MB}–F+H_{2}O$$$$2\mathrm{MB}–\mathrm{OH}\;+2F^{-}\;↔\;2\mathrm{MB}–F+2{\mathrm{OH}}^{-}$$
where MB stands for magnetic biochar.

Additionally, the mean adsorption energy (*E*) calculated by the D-R isotherm (1.06 kj/mol–0.90 kj/mol) suggests that physisorption is the primary step in adsorption. The ∆*H^o^* values show the exothermic process that was controlled by physical adsorption [30]. The ∆*G^o^* values obtained in this study show that the adsorption process was spontaneous and favorable at low temperatures. The XRD and FTIR data revealed the formation of new peaks after fluoride adsorption, confirming the formation of a unique bond between the carboxylic group Fe and F, while the BET data confirm the existence of a large surface area and pore volume and diameters with a mesoporous structure. The fluoride adsorption mechanism on the MTBC surface is displayed in Figure 12.

### 3.10. Economic Implication for MTBC Usage in the Real World

The importance of this sorbent is based on its low-cost values, which helpfully remove fluoride from water. So, the cost analysis during this study was investigated. A stepwise cost analysis (in USD) is described as follows:Raw material = 0 USD (collected from different cafes and teashops).Sample processing = 0.055 USD (tap water was used for initial wash. Distilled water was only used for rinsing [0.5 l × 0.11 USD per liter cost]).Fe (NO_3_)_3_ impregnation = 0.148 [5.5 g of Fe (NO_3_)_3_ was dissolved in 50 mL of distilled water for 10 g tea feedstock (5.5 × 0.027 cost per gram)].Pyrolysis = 0.516 USD (6 Units × 0.086 cost per unit).Total cost per gram = 0.0719 USD (0.719 ÷ 10).Total cost per Kg = 71.9 USD.

Many countries contain high fluoride levels in their drinking water, especially Pakistan, India, China, USA, Italy, Ghana, Mexico, Sudan, Nigeria, etc. [53]. Developed countries can afford high-price methods, such as ion exchange, nanofiltration, and precipitation. However, developing countries need low-cost techniques with high removal efficiency to remove the fluoride from water.

## 4. Conclusions

The one-step pyrolysis process was used to prepare the TBC, which was impregnated by Fe (NO_3_)_3_ to form MTBC and successfully applied in groundwater defluoridation. The surface area of TBC (81.64 m^2^/g) was smaller than MTBC (115.65 m^2^/g). TBC and MTBC were characterized by their main components, carbon and oxygen along with other trace elements found in MTBC. The oxygen content of MTBC (25.56 wt %) was lower than that of TBC (26.18 wt %). MTBC was used to remove fluoride, and its removal efficiency reached 98% for the initial fluoride concentration of 50 mg/L at optimized conditions (pH: 3.6, temperature: 308 K, and time: 24 h). The adsorption kinetics data explained the adsorption carried out by the chemical process and best fit the pseudo 2nd order model. The adsorption data are better fitted to the Freundlich model rather than the Langmuir model, which described the fluoride adsorption process controlled by multilayered process. The competitive ions Cl^−^ and HCO_3_^−^ have a slight effect, while other competitive ions (NO_3_^−^, CO_3_^2−^ and SO_4_^2−^) have a more significant impact on fluoride adsorption, due to their high charge density. Ion exchange, inner-sphere complex, electrostatic force of attraction, and hydrogen bonding probably take part in the defluoridation. Regeneration study and actual water treatment reveal that this adsorbent is helpful and supportable in fluoride removal in commercial and industrial uses because of its low cost and high removal efficiency.

## Figures and Tables

**Figure 1 ijerph-19-13092-f001:**
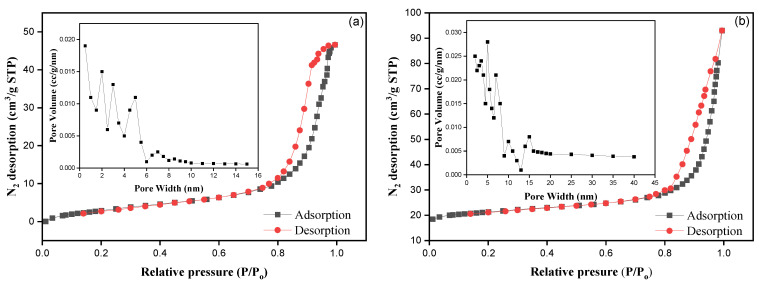
(**a**) N_2_ adsorption–desorption curves of TBC, along with pore volume; (**b**) N_2_ adsorption–desorption curves of MTBC, along with pore volume.

**Figure 2 ijerph-19-13092-f002:**
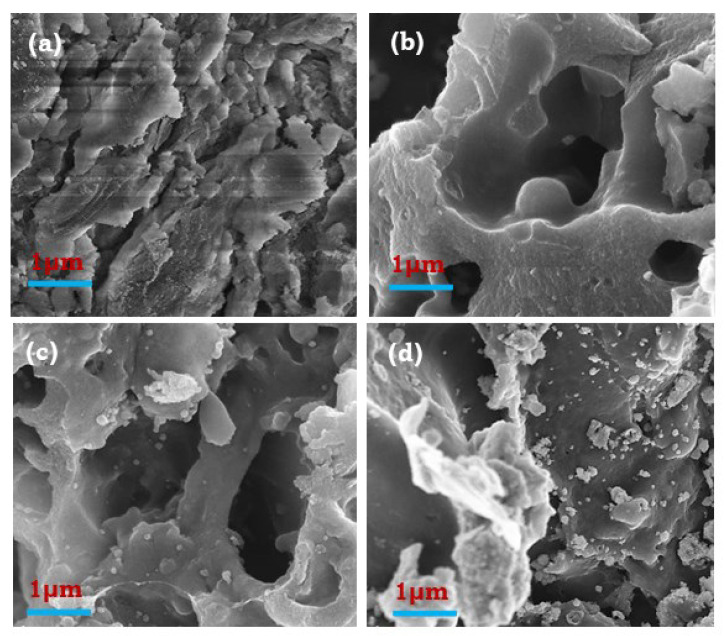
SEM images of (**a**) tea waste; (**b**) TBC; (**c**) MTBC; (**d**) MTBC-F.

**Figure 3 ijerph-19-13092-f003:**
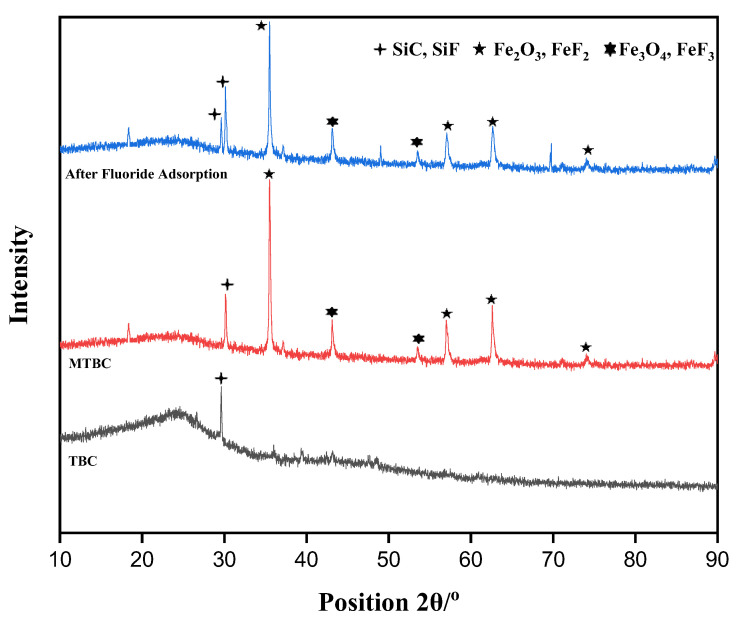
XRD pattern of TBC, MTBC, and MTBC-F.

**Figure 4 ijerph-19-13092-f004:**
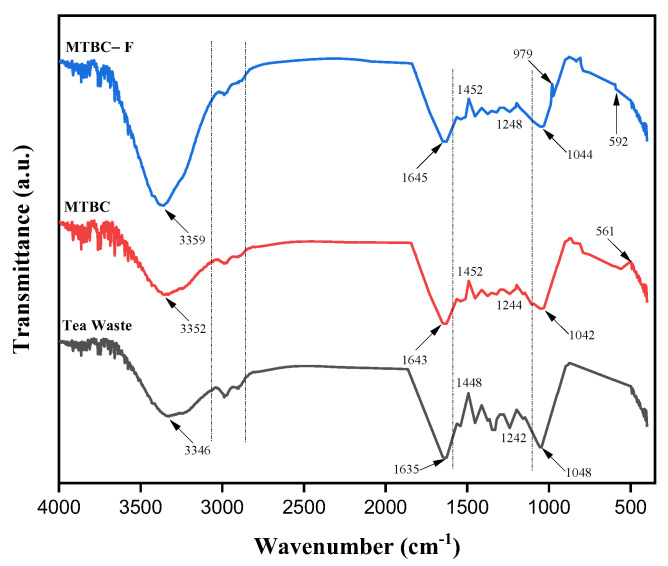
FTIR spectra of tea waste, MTBC, and MTBC–F.

**Figure 5 ijerph-19-13092-f005:**
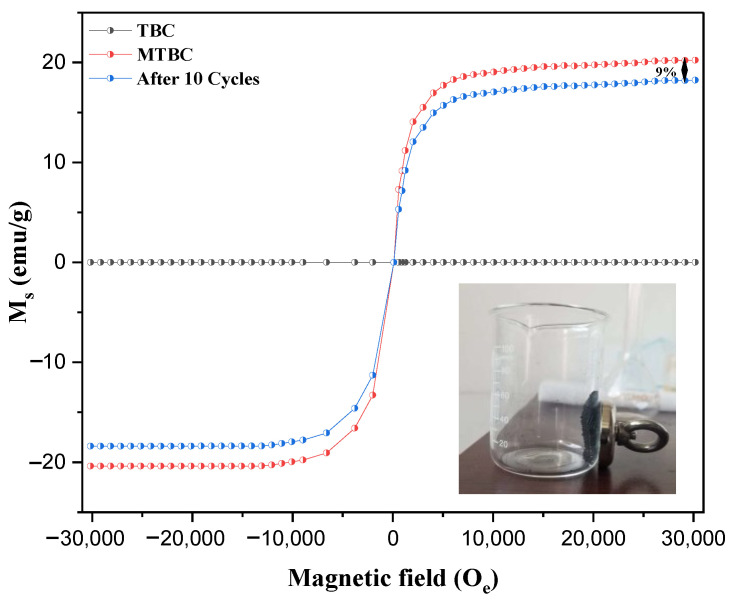
VSM of TBC, MTBC, and after 10 reused cycles.

**Figure 6 ijerph-19-13092-f006:**
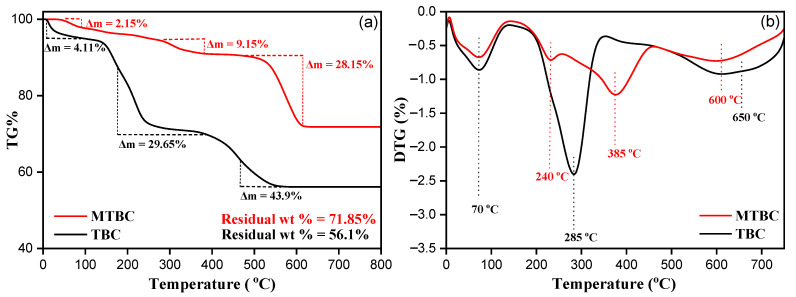
(**a**) TG and (**b**) DTG analysis of TBC and MTBC.

**Figure 7 ijerph-19-13092-f007:**
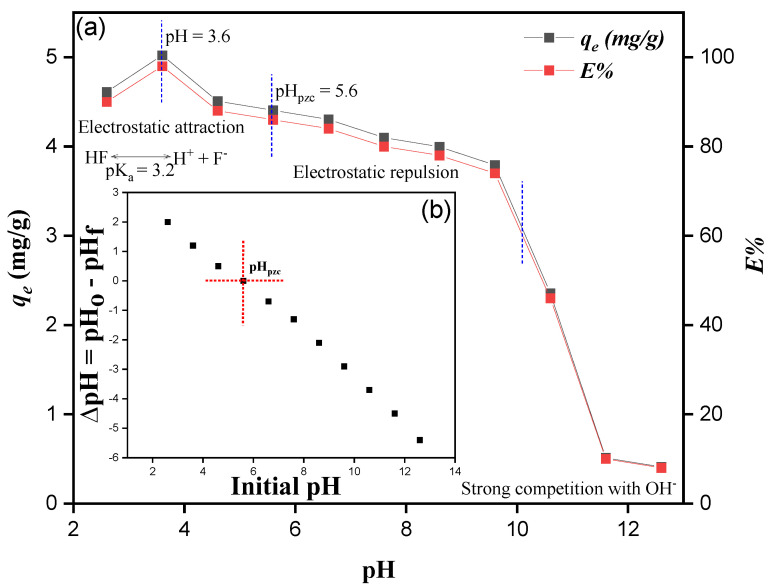
(**a**) Effect of pH and (**b**) pH_PZC_ values of MTBC.

**Figure 8 ijerph-19-13092-f008:**
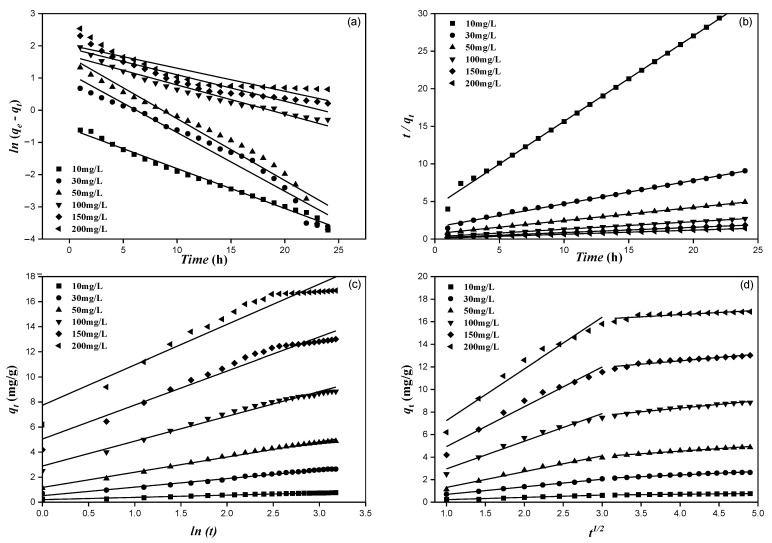
Adsorption kinetic models (**a**) Pseudo 1st order; (**b**) Pseudo 2nd order; (**c**) Elovich; (**d**) Intraparticle diffusion.

**Figure 9 ijerph-19-13092-f009:**
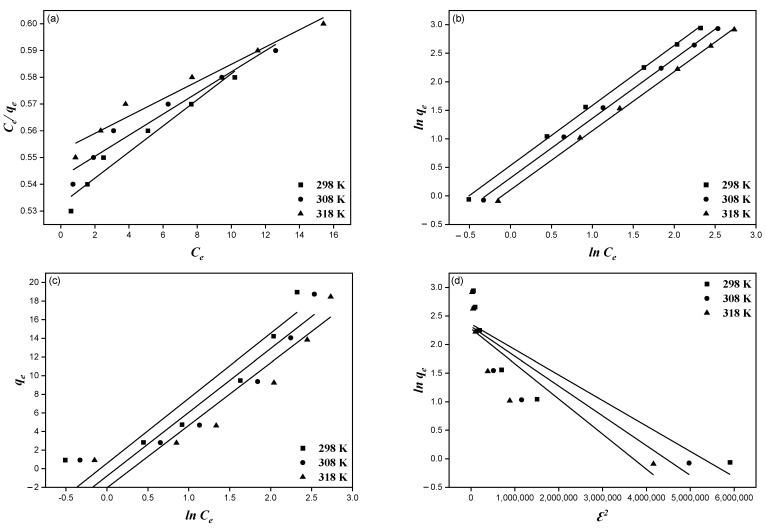
Adsorption isotherm models (**a**) Langmuir; (**b**) Freundlich; (**c**) Temkin; (**d**) Dubnin-Radushkevich.

**Figure 10 ijerph-19-13092-f010:**
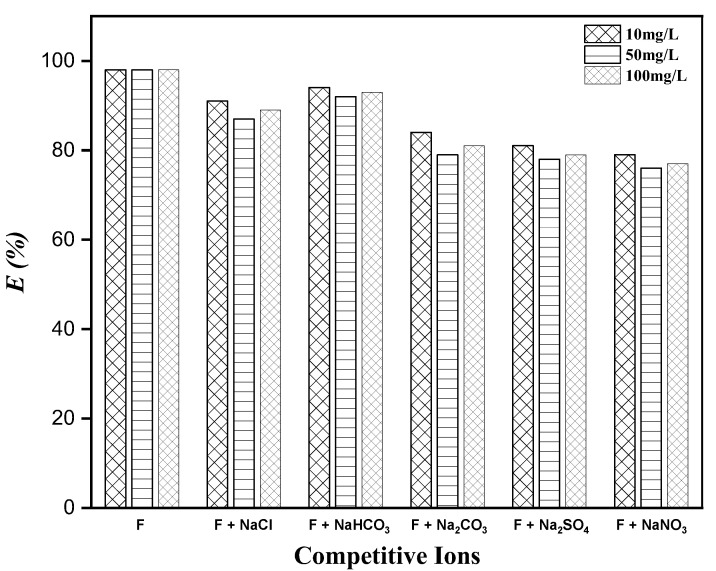
Effect of competitive ions’ presence in drinking water.

**Figure 11 ijerph-19-13092-f011:**
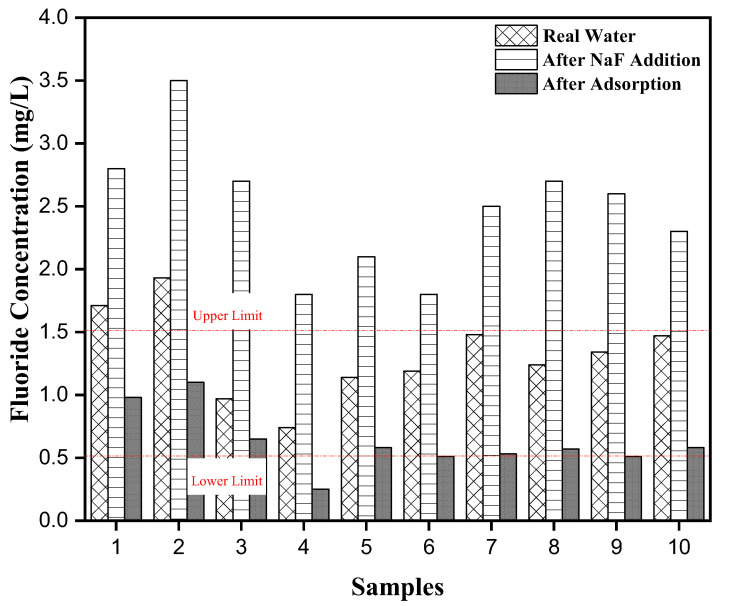
Groundwater defluoridation by using MTBC.

**Figure 12 ijerph-19-13092-f012:**
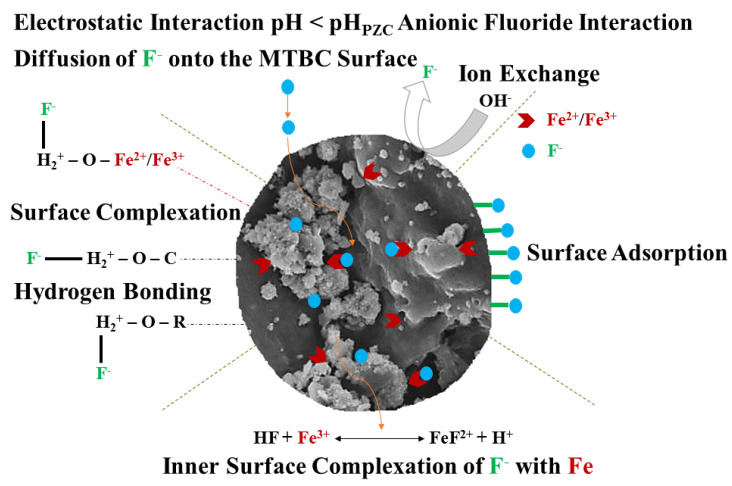
Proposed mechanisms on the MTBC surface.

**Table 1 ijerph-19-13092-t001:** Proximate and ultimate analysis of TBC and MTBC.

Samples	Proximate Analysis				Ultimate Analysis					BET Analysis			pH
	*MC*	*AC*	*VM*	*FC*	*%C*	*%H*	*%N*	*%O*	Other	S_BET_ (m^2^/g)	Pore Volume (cm^3^/g)	Average Pore Width (nm)	
**TBC**	1.96	5.84	74.56	17.64	63.45	4.52	5.46	26.18	0.39	81.64	0.21	2.06	5.94
**MTBC**	1.86	5.34	79.73	13.07	63.75	4.65	5.63	25.56	0.41	115.65	0.40	5.42	5.78

**Table 2 ijerph-19-13092-t002:** Kinetic model parameters for fluoride adsorption on MTBC.

*C_o_* (mg/L)	*q_e,exp_* (mg/g)	Pseudo 1st Order			Pseudo 2nd Order			
		*q_e_, Cal* (mg/g)	*k_1_*_× 10_^−3^ (h^−1^)	r_1_^2^	*q_e,cal_* (mg/g)	*k_2_* (g/(mg·h)	*h* (mg/g·h)	r_2_^2^
10 30	0.79	0.56	−5.2	0.9912	0.88	0.30	0.23	0.9979
	2.69	3.11	−7.6	0.9564	3.21	0.06	0.64	0.9966
50	4.90	5.24	−8.0	0.9559	5.72	0.04	1.41	0.9999
100	9.60	5.43	−3.8	0.9573	9.97	0.03	3.36	0.9999
150	14.25	6.84	−3.4	0.8913	14.34	0.03	6.26	0.9994
200	18.79	7.55	−3.0	0.7918	18.29	0.03	10.91	0.9989
	**Elovich model**			**Intraparticle diffusion model**				
***C_o_* (mg/L)**	α **(g/mg·h)**	***β* (g/mg·h)**	**r_E_^2^**	***k_d_1* (mg/g·h^1/2^)**		***k_d_2* (mg/g·h^1/2^)**	** *C* **	**ri^2^**
10	5.53	−0.57	0.9802	0.20		0.07	0.23	0.9765
30	1.47	0.38	0.9881	0.67		0.31	0.61	0.9980
50	0.82	1.17	0.9932	1.40		0.46	1.47	0.9823
100	0.50	2.14	0.9833	2.45		0.66	3.12	0.9683
150	0.37	2.86	0.9622	3.53		0.58	5.81	0.9622
200	0.31	3.57	0.9268	4.58		0.38	8.88	0.9544

**Table 3 ijerph-19-13092-t003:** Intraparticle diffusion parameters for fluoride adsorption on MTBC.

*C_o_* (mg/L)	*k_d_1* (mg/g·h^1/2^)	*C*	*Ri*	*Ri* Description		Adsorption Behavior
				*Ri* = 1	No Initial Adsorption	
10	0.20	0.02	0.97	0.9 < *Ri* < 1	Weak initial adsorption	Weak initial adsorption
30	0.67	0.02	0.99			Weak initial adsorption
50	1.40	0.26	0.90	0.5 < *Ri* < 0.9	Intermediate initial adsorption	Intermediate initial adsorption
100	2.45	0.50	0.89			Intermediate initial adsorption
150	3.53	1.41	0.87	0.1 < *Ri* < 0.5	Strong initial adsorption	Intermediate initial adsorption
200	4.58	2.67	0.86			Intermediate initial adsorption
				*Ri* < 0.1	Adsorption completed in a short time	

**Table 4 ijerph-19-13092-t004:** Thermodynamic model parameters for fluoride adsorption on MTBC.

Models	T (K)	Parameters			
		*R_L_*	*K_L_* (Lmg^−1^)	*q_mL_* (mg/g)	r^2^
Langmuir	298	0.89	1.09	18.78	0.9503
	308	0.90	0.91	18.43	0.9513
	318	0.91	0.89	18.10	0.9507
		*K_F_* (Lmg^−1^)	*N*		r^2^
Freundlich	298	1.71	2.30		0.9979
	308	1.37	2.34		0.9986
	318	1.11	2.39		0.9981
		*K_T_* (Lmg^−1^)	*b* (kjmol^−1^)		r^2^
Temkin	298	1.09	6.98		0.8297
	308	0.89	6.85		0.8252
	318	0.74	6.71		0.8183
		*b_DR_* _× 10_ ^−7^	*E* (kjmol^−1^)	*q_mDR_* (mg/g)	r^2^
Dubnin–Radushkevich	298	4.46	1.06	10.59	0.7655
	308	5.21	0.98	10.15	0.7438
	318	6.15	0.90	9.77	0.7256

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
