# Peer review of "Nanoarchitectonics and Kinetics Insights into Fluoride Removal from Drinking Water Using Magnetic Tea Biochar"

_ijerph, 2022, doi:10.3390/ijerph192013092_

Round 1
Reviewer 1 Report
The authors have studied a low-cost magnetic tea biochar (MTBC), prepared by facile one-step pyrolysis of waste tea leaves, in order to remove fluoride contamination from natural water. MTBC product was characterized by XRD, SEM, FTIR, and VSM. In optimum conditions (pH 3.6, Temperature 308 K, and time 24 h), the MTBC reached 98% removal efficiency for 50 mg/L initial fluoride concentration.
To be accepted for publishing, the manuscript should be rewritten, in order to improve its clarity, structure and attractivity. Also, the authors should pay more attention on typos, English language and text format (use journal’s template).
Supplementary, some revisions are necessary:
1. At page 1, Abstract: Please add measurement units of surface area for both, TBC and MTBC.
2. At pages 6-7, section 3.2.1. and Figure 2(a-d): How the authors determined/ put in evidence the Fe particles (2c) and fluoride (2d), without SEM-EDX analyses? I recommend them to perform SEM-EDX analyses.
3. At pages 7, Figure 2(a-d): The scale bar of SEM images are not visible. How the authors compared BET results (2-5 nm average pore width) with SEM images (2 µm scale bar)?
4. At page 13, Figure 8 legend: Please add the (d) kinetic model type.
5. At page 17: The scheme of ion-exchange mechanism must be rewritten.
6. At page 17, Figure 12: The quality of the scheme must be improved.
Author Response
All suggested modifications have been done according to the editorial and reviewers' comments.

Reviewer 2 Report
Although this manuscript included lots of insufficient aspects, it actually presents interesting data. Publication of these data in public journal media would have good contributions to the related research fields. From the latter positive viewpoint, I may recommend publication of this work in Environmental Science and Engineering. However, several revisions are necessary. Please see below.
1) In this manuscript, font sizes are not well unified. Please use the same font size.
2) In such application-inclusive research, comparisons over the other systems reported in the past literatures are important. Actually the authors represents Table S1 for comparisons. However, the authors simply described … However, it is impossible to conclude adsorbent capacities, as these were calculated at different pH, temperature, adsorbent dosage, adsorbent concentration range, adsorbent particle size, and surface area … It is true. However, the authors had better say something insightful consideration with general tendencies with these data.
3) In the title, Experimental … is too broad and virtually meaningless. This work is based on consideration with nanoarchitectures and kinetic behaviors. Therefore, I may recommend to change the title like … Nanoarchitectonics and kinetics insights into fluoride removal from drinking water using magnetic tea–biochar … may sound more reasonable.
4) In page 17, equations look disordered (arrows are not well arranged). Please check the pdf rile of your manuscript.
5) Reference are not bad but can be updated and generalized. In addition, detection of fluoride is important backgrounds. It may be better to add short descriptions on fluoride detection with citing recent papers (for example, https://www.journal.csj.jp/doi/abs/10.1246/bcsj.20200003, https://pubs.rsc.org/en/content/articlelanding/2022/NJ/D2NJ01946A).
6) In Figure 2, clear scale bars and scales have to be added to images in Figure 2.
7) In Figure 4, the vertical axis parameter should be Transmittance (a.u.) (not T%) because the axis does not have values and % cannot be used.
Author Response

(The authors gave the same response as above.)

Reviewer 3 Report
Reviewer comments: The article is interesting, results are valuable, however, some parts of manuscript are not clear. Also, there are numerous typos in the manuscript. Furthermore, some parts of the manuscript have to be improved. 1. Please provide more information on the preparation of TBC, reported preparation is not clear and needs more discussion of the science 2. Please provide more information on the preparation of the real water sample, reported preparation has no information on dilution 3. What is the mechanism for the adsorption? 4. What is the nature of the magnetic particles in terms of hydrophobicity? 5. What is the ration of production from? 6. SEM images are unclear and their scale bar is not clear as well. 7. The manuscript needs major revision for English including, sentence structures and front formatting. 8. The photographs are not in a high resolution, and not consistent. 9. According to your studies its better to refer this articles as well. https://doi.org/10.3390/su122410646
Author Response

(The authors gave the same response as above.)

Round 2
Reviewer 1 Report
The authors performed some modifications suggested by reviewers and improved their manuscript. However, the revised version of the manuscript needs further improvements (in terms of structure and attractivity) in order to be recommended for publication. Supplementary, there are still necessary some particular revisions:
1. In Figure 2 (a-d) is still missing the scale bars. Please add them.
2. Please show (explain) how "Figure (2c) describes [...] the attachment of Fe particles". Could you evidenced Fe particles on Figure 2c?
Author Response
Dear Editors, all suggested modifications have been done according to the editorial and reviewers' comments. All the mentioned issues in the reviewers' comments are well addressed and discussed with practical reasons, and the red-colored font highlights every change. Now we have a great hope that potential reviewers will accept our Manuscript for publication.

Reviewer 2 Report
Replies and revisions are fine. The revised version becomes acceptable.
Author Response
Dear Editors, all suggested modifications have been done according to the editorial and reviewers' comments.